# Risk Mitigation and Investability of a U-PHS Project in The Netherlands

**Gert Jan Kramer** [1,*] , **Twan Arts** [2] , **Janos L. Urai** [3,4] , **Han Vrijling** [5] **and Jan M. H. Huynen** [6]

1 Copernicus Institute of Sustainable Development, Utrecht University, Princetonlaan 8a, 3584 CB Utrecht, The Netherlands

2 O-PAC Ontwikkelingsmij, Vrijthof 48, 6211 LE Maastricht, The Netherlands; twanarts@o-pac.nl

3 Institute for Structural Geology, Tectonics and Geomechanics, RWTH Aachen University, Lochnerstrasse 4-20, D-52056 Aachen, Germany; janos.urai@geostructures.nl

4 Geostructures—Consultancy for Structural Geology and Geomechanics, Hunnenweg 9, 6224 JN Maastricht, The Netherlands

5 Department of Hydraulic Engineering, Delft University of Technology, Stevinweg 1, 2628 CN Delft, The Netherlands; j.k.vrijling@tudelft.nl

6 Sogecom B.V., Vrijthof 48, 6211 LE Maastricht, The Netherlands; jmh@sogecom.nl

\* Correspondence: g.j.kramer@uu.nl; Tel.: +31-30-253-7948

**Abstract:** We review the status of a 1.4 GW, 8 GWh underground pumped hydro storage (U-PHS) project in the southern Netherlands, which has been under development since the 1980s. Its history shows how the prospect of a large-scale U-PHS for The Netherlands (a country whose proverbial flatness prohibits PHS) has been attractive in every decade, based on proven technology in a subsurface location with validated properties, and solid analysis of its economics. Although the ongoing energy transition clearly requires massive electricity storage, (U-)PHS projects are challenging investment propositions, in The Netherlands, as elsewhere. This case study illustrates a point of general relevance, namely that although the project execution risk, related to uncertainty with respect to subsurface integrity, is very low, the transition risk, associated with the intrinsic uncertainties of an electricity system in transition, is significant. We point out mitigation strategies for both risk categories.

**Keywords:** pumped hydro storage; electricity storage; renewable energy; electric grid stabilization; energy transition; regional economic development

## 1. Introduction and Project History

Pumped hydro storage (PHS) is the dominant technology to store energy. It uses a water reservoir at elevated heights to store a surplus of electricity supply and releases it to generate electricity when there is a supply shortage. PHS accounts for 95% of the 191 GW grid-connected storage globally [1]. The Netherlands and Denmark are the only countries in Western Europe without significant relief. As a consequence, storage based on conventional PHS is not an option.

Underground pumped hydro storage or U-PHS [2], illustrated in Figure 1, would therefore seem to be an option of obvious interest to The Netherlands [3]. The country has a well-known subsurface with a layer of strong and stable limestone of Dinantian age in the Southern Netherlands at a depth of 1000–1500 m, eminently suitable for U-PHS projects. In this paper, we pose the question of why it has proven so difficult to realize the first Dutch U-PHS project, a project that has been on the drawing board since the 1980s and that, in each subsequent decade, was shown to be an attractive addition to the Dutch and regional (Dutch, Belgium, German) electricity infrastructure.

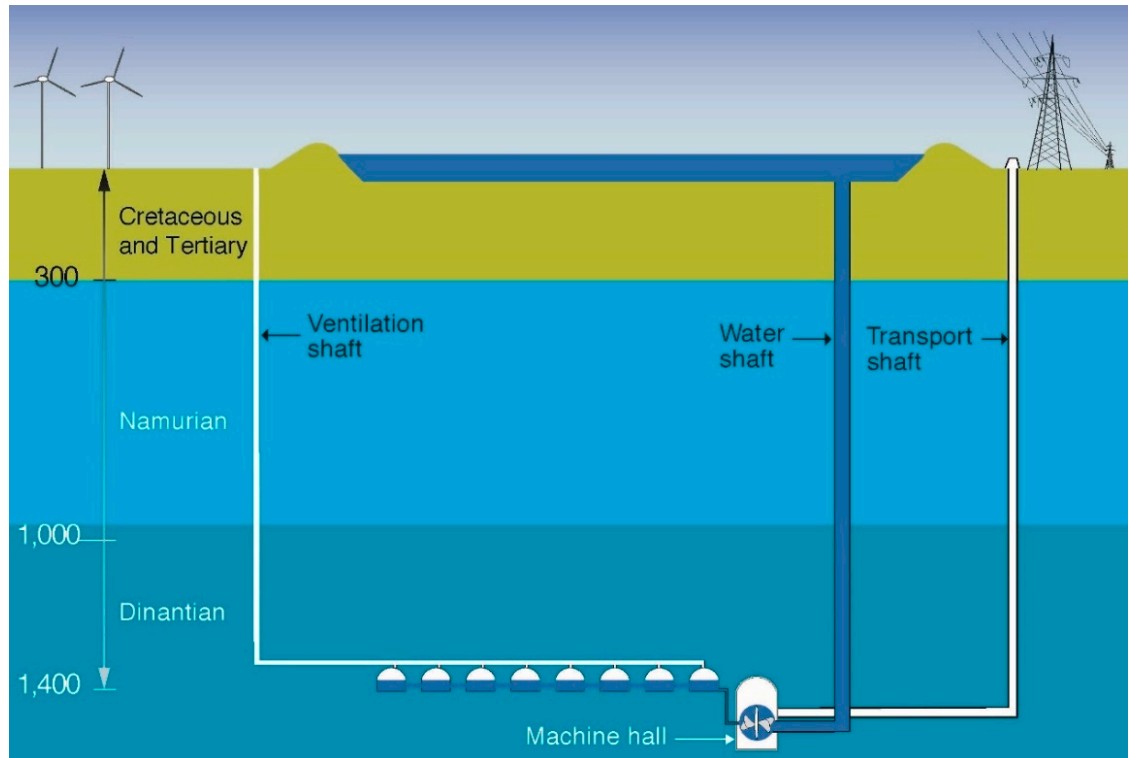

**Figure 1.** Schematic profile (only approximately to scale) of the O-PAC underground pumped hydro storage (U-PHS) in the area around Geverik, Limburg, showing a simplified geologic structure with Dinantian Kolenkalk formation below about 1000 m. The lower reservoir consists of a system of tunnels about 12 km long with approximately 16 m diameter, distributed over an area of about 500 × 1000 m.

The history of O-PAC, an acronym for *Ondergrondse Pomp Accumulatie Centrale* (Dutch for underground pump accumulation plant), a 1.4 GW, 8 GWh, 2 billion € project plan for U-PHS in the province of Limburg in the south of The Netherlands, is reviewed in Section 2. In Section 3 we review the project-specific geologic and geotechnical information in the Southern Netherlands and show that the presence of suitable rock formation at the required depth has been convincingly demonstrated, and in addition cost of construction can be minimized by choosing a location with minimal karst and natural fractures. In Section 4 we present a benefit/cost analysis of the project under current market regulations, but with a power supply portfolio that increasingly relies on intermittent renewables, as per the projections of the Dutch *Klimaatakkoord* [4], this reveals that only half of the financial benefit is reaped by the O-PAC owner; the other half is societal benefit that accrues to consumers as lower (average) electricity price, to transmission system operators through grid stabilization, and to owners of renewable generation in the form of higher prices at times of renewable oversupply.

These three sections show that as a business proposition, O-PAC is a low-risk, low-return project. However, considered as an infrastructure investment, the project is attractive and has a good monetary return provided the downside risk of change in market regulation is covered. Moreover, the project has considerable, but less tangible, non-financial benefits by making the Dutch and regional electricity system more resilient. From a technical and techno-economic perspective, O-PAC has characteristics which make it complementary to the more fashionable modes of supply–demand management, viz. battery storage and demand-side management. This is discussed in Section 5, which is more conjectural in nature, but sets us up for the conclusion in Section 6, where we will argue that the investment potential of O-PAC, and by extension any U-PHS, or indeed any large-scale storage project, is hampered by it being high-risk-low-return from a private investor perspective, while being low-risk-good-return from a societal perspective. This points to market failure and the critical role

that government will have to play in creating the conditions under which U-PHS becomes investable through public–private partnership.

## 2. History and Main Characteristics of O-PAC

Because of the global dominance of PHS for electricity storage and the absence of suitable topography in The Netherlands, over forty years ago the first suggestions had already been made to use subsurface reservoirs for PHS, creating an underground PHS (U-PHS). At the time, the Dutch government and the public-owned utility companies saw an emerging need for storage to optimize their electricity production portfolio, as large-scale storage would allow simultaneously the maximization of the utilization of base load capacity (coal and nuclear) and the minimization of the use of peak load capacity (gas and oil) [5].

The question was raised whether the then recently-closed coal mines in Limburg (The Netherlands' southernmost province, bordering on Belgium and Germany) could be used as reservoirs. In 1980, the mining engineering department of TH Delft (now Delft University of Technology) undertook a study into this option but concluded that the caved-in, abandoned mines in Limburg would be difficult to repurpose as a lower basin for a U-PHS [6]. In line with this, a more recent project to explore the prospects of U-PHS in abandoned coalmines in the Ruhr Region in Germany concluded that creating a new, purpose-built lower reservoir was to be preferred to using the partly collapsed old mine workings [7,8].

In 1983, the Dutch parliament asked the Minister of Economic Affairs to start a feasibility study of large-scale energy storage. In response, an OPAC group was formed and it conceived a plan for a subterranean PHS project based on a high head and with an output of 1400 MW. The OPAC project team initially consisted of TH Delft, the engineering consultancy firm Haskoning, Volker Stevin, a construction company, Aveco, an infrastructure consultant, and Sogecom, a development company. The Ministry of Economic Affairs appointed the then energy development agency NEOM (*Nederlandse Energie Ontwikkelings Maatschappij*) as coordinator of the study. Due to the specialized knowledge required for tunneling and cavern construction, the German engineering firms Müller–Hereth, Deilmann–Haniel, and Heitkamp joined the study group and have continued to contribute their expertise to the present day. The study covered all technical aspects, from engineering to construction and operation.

U-PHS requires a rock formation which is strong and stable enough to allow construction of large underground facilities at a depth of 1000–1500 m. The head must be as large as possible in order to minimize cavern volume and thereby costs; the limit is set by the head considered feasible by pump-turbine manufacturers, about 1400 m. This large head has the additional advantage that dynamic variations are minimal. These conditions are fulfilled in the south of The Netherlands. The original OPAC study assessed environmental constraints, licenses, permits, and financial feasibility. During these studies, building on extensive geologic knowledge of the coal mines and surface outcrops of massive Dinantian limestones further south, drill core samples were taken in well GVK-1 in South Limburg, and a 2D seismic investigation was executed to identify the main structures of the subsurface. Analysis by the *Rijks Geologische Dienst* (RGD, then the national geologic survey) and the Laboratory of Engineering Geology of TH Delft confirmed the suitability of the subsurface around GVK-1 for a U-PHS. The Dinantian Kolenkalk layers between approximately 1000 and 1700 m depth offer a stable rock formation to host U-PHS [9]. In Section 3 we elaborate on these findings.

NEOM commissioned Motor Columbus, a Swiss advisory engineering group, to validate the results [5]. The study concluded that the optimal depth for a U-PHS unit at the designated site in Limburg was 1400 m, and recommended 1400 MW and 8.4 GWh for power and storage capacity, respectively (see Figure 1).

The original, 1980s OPAC project delivered a complete, investment-grade project plan for the development and construction of a U-PHS. In 1989, and in spite of this positive technical assessment, the Dutch government chose to not support investment in the plan of the OPAC study team. One reason for the hesitation and suspension of the plan was the sheer size of the project, which required a 2528 Mfl.

investment (ca. 1930 M€ in today's money). The other was that at that same time, plans for additional nuclear power plants were also postponed, plans that had initially been an important part of the rationale for large-scale storage (as nuclear plants are best operated continuously). Additionally, since large quantities of natural gas were cheaply available, flexibility of the power supply was not an urgent concern.

Some twenty years later, Sogecom, a development company, revived the project under the name of O-PAC. (The hyphen in its acronym distinguishes this resumption from its predecessor OPAC). In anticipation of an increasing share of intermittent renewables and decentralized power production and with the support of the Province of Limburg, O-PAC updated the existing plans and presented a project plan, a business case, and an assessment of regional economic benefits. The energy companies Essent, Nuon, and E.ON Wasserkraft were prepared to invest € 1 billion, under the condition that the government would provide a particular financial guarantee for the first years of operation. The Ministry of Economic Affairs declined to do so because it was reluctant to interfere in the market and because it was at that time not convinced of the necessity of storage. In order to re-assess the subsurface feasibility of O-PAC in Southern Limburg, the Province of Limburg commissioned the TNO (Netherlands Organization for Applied Scientific Research) and Geostructures, an independent geological consultant, to study the geological situation in 2011 [10].

Since 2010, the view that intermittent renewables will become the dominant source of power has become fully mainstream. In conjunction with that, the need for electricity storage is also uncontested. Today only 12% of Dutch electricity comes from solar and wind, set to grow to 70% by 2030 according to the plans laid down in the Climate Agreement of 2019. This agreement does not explicitly address storage. The familiar range of options to deal with intermittency is mentioned: demand response, grid expansion alongside with storage, for which there is special mention of hydrogen, but the investment and policy focus is very much on generation [4].

It is in this context that we review the present case for O-PAC below; geology (Section 3), business case (Section 4), and outlook for investment (Section 5).

## 3. Geological Risk and Its Mitigation—Subsurface Aspects of U-PHS in Limburg

There are two fundamentally different kinds of anthropogenic activity in the deep subsurface. The first is related to the mining of minerals, the production of hydrocarbons, or geothermal energy. These activities can lead to land subsidence or induced seismicity, of which several examples are known in The Netherlands [11]. The second is related to the creation of stable underground construction for tunnels or shafts. These are designed and built for long-term stability and operation, with minimal subsidence, very high subsurface integrity, and resistance to natural earthquakes [12]. Examples are tunnels or pumped hydro power stations.

The O-PAC underground pumped hydro storage (U-PHS) facility belongs to the second kind. Its elements, shafts, machine hall, and tunnels, are based on proven technology and high standards of long-term stability, used worldwide. Underground construction is common at shallow depth in The Netherlands, but construction at 1400 m depth would be new to the country.

The construction of an underground water reservoir in tunnels and the large machine hall containing the hydroelectric installations requires a strong, stable rock formation. Plans for the subsurface engineering for O-PAC in the south of Limburg envisage an upper reservoir at surface of ca. 500 × 500 m, connected to an underground reservoir and machine hall at 1400 m depth with three shafts (see Figure 1). The underground reservoir and machine hall will be constructed in a thick, tectonically largely undisturbed package of silicified limestones. These rocks, of the lower carboniferous (Dinantian) age Kolenkalk formation, have been buried to over 5000 m depth in their geological history, which resulted in compaction, very low permeability, and high cohesion [13].

The present knowledge base for the subsurface O-PAC is based on investigations which started about 100 years ago in relation to coal mining [3]. The coal mining era, which in Limburg lasted until the 1970s, produced an extensive dataset of the area to the north and northeast of the potential locations

for O-PAC. Although these investigations focused on the coal-bearing upper Carboniferous layers, the laterally rather continuous layer thicknesses, together with a gentle dip of the layers towards the North, allowed projection of the top of the Dinantian Kolenkalk formation, and the Paleozoic subcrop map below the cover sediments provided a clear picture of the geologic structure. This was summarized by Dikkers [14] and Sax [15], who clearly separated the gentle folding and faulting of the Variscan fold-and-thrust belt exposed in the Ardennes, and the normal faulting of the Ruhr Valley Graben, which has been active until the present with associated seismicity. The coal mining also produced a very high-resolution map of faults and fractures which can be extrapolated to the target area to predict sub-seismic fractures and faults, and help optimize the precise location and orientation of the lower reservoir and machine hall of O-PAC.

The subsurface part of the O-PAC project consisted of the drilling, logging, and coring of the borehole Geverik-1 and the acquisition and interpretation of a number of 2D seismic lines. The location of the Geverik borehole was chosen based on the tectonically quiet location at the NE-tip of the Brabant Massif, where the karstification of the top Dinantian as observed in other areas was expected to be absent, and Dinantian limestones were predicted to be present at depth below 1000 m (Figures 2 and 3).

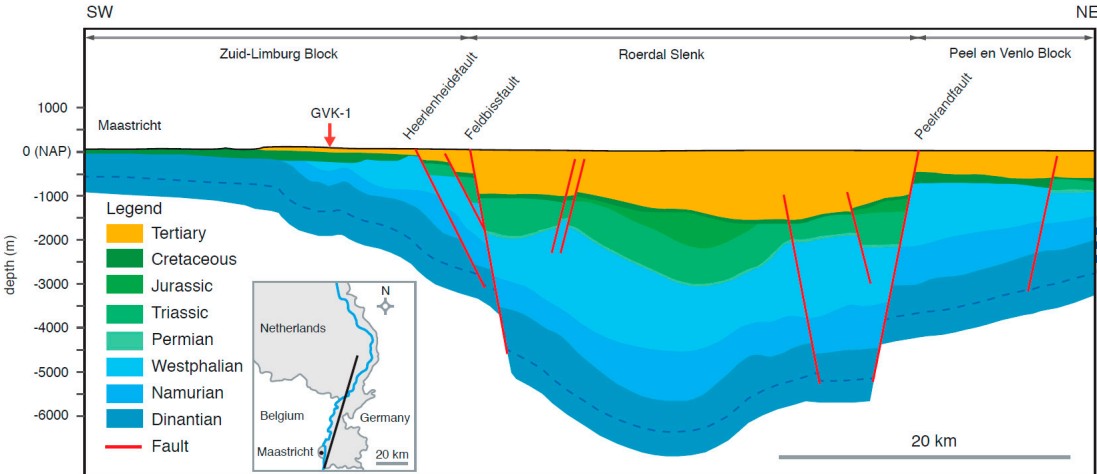

**Figure 2.** Regional profile through Limburg, showing the main tectonic units and stratigraphy. Note that the well GVK-1 in Geverik did not reach the bottom of the Dinantian (Kolenkalk) limestones, so the base of this unit is not well known. In fact, the well only penetrated the upper part of the Dinantian (the Visean). The dashed blue line shows the inferred depth where presence of Dinantian is based on data, below this line we show possible deeper occurrence. The O-PAC area in the vicinity of the borehole GVK-1 is characterized by layers gently dipping to the NE, with minor faults and fractures. After [16].

Results of the seismic, and especially the very extensive Geverik-1, core investigation confirmed these predictions [17]. Below 1000 m depth, the interbedded, partly dolomitized and silicified limestones of the Dinantian Kolenkalk formation (Carboniferous Limestone Group) were drilled for almost 700 m, with a gentle dip towards the NE, intersecting a number of small-offset faults and joints. The strength of the compacted and cemented limestones varies from strong (i.e., having a uniaxial compressive strength (UCS) greater than 100 MPa) to extremely strong, providing the desired rock mass for the O-PAC project. Karstification was shown to be absent in Geverik-1 [18]. In addition, the very extensive geotechnical investigations and extensive characterization of the rock discontinuities [19] provided a large database, which forms an excellent basis (by today's standards) for a model of the rock mass (Figure 4).

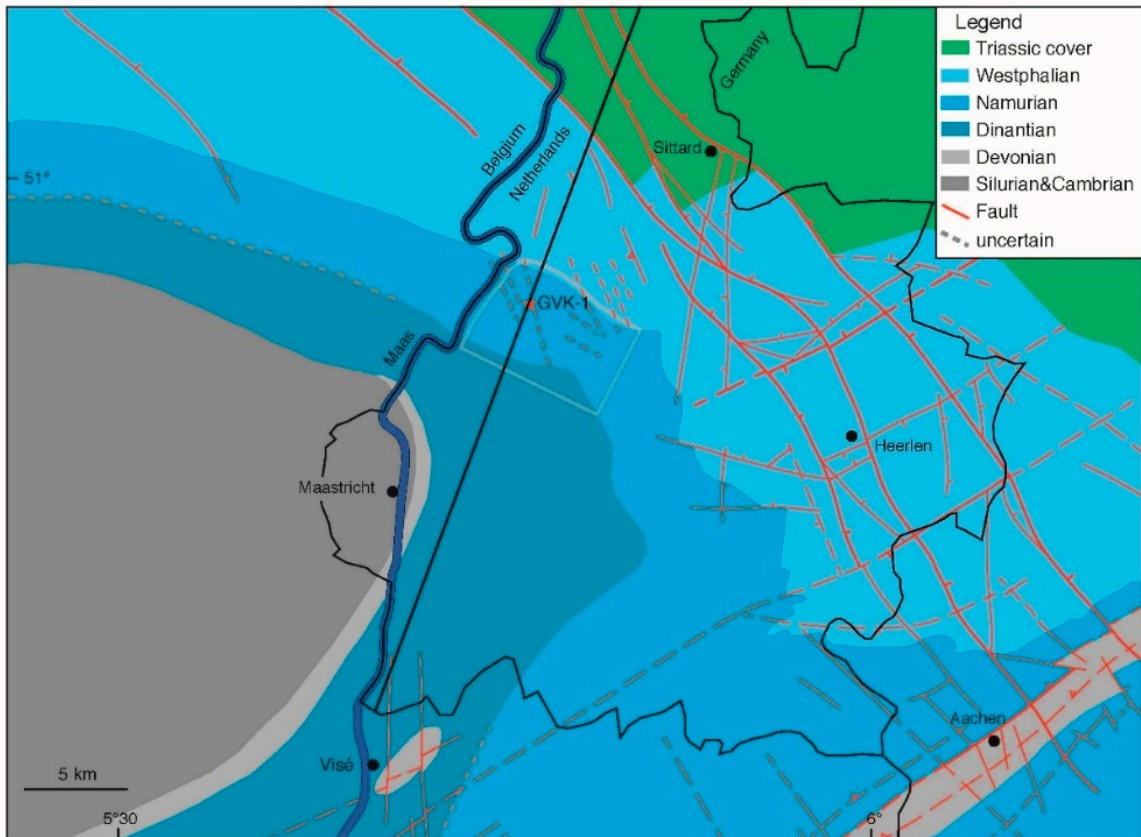

**Figure 3.** Geologic map of the top Paleozoic erosion surface (i.e., the cover layers in Figure 2 removed) showing the relatively simple structure of the Carboniferous layers on the NE side of the Brabant Massif, with the location of the GVK-1 well in gently dipping Carboniferous strata at a distance from both the Variscan thrust front in the SE near Aachen, and the Ruhr valley graben faults in the NE which are active at present. Small faults (such as the Geulle fault) which are known in the area and were drilled in the GVK-1 well are shown schematically by broken lines. The thin yellow line shows the area which is suitable for the lower reservoir of U-PHS. The straight black line is the same profile line as in Figure 2.

These results were summarized in Müller and Hereth [9], and used for the design of the subsurface engineering works. The conclusion of this study was that, based on these results and using technology of 1987, building the underground reservoir for O-PAC was feasible. The strength, homogeneity and the very low permeability of the rock ensure that the basic requirements for such a construction were fulfilled.

More recently, the report commissioned by the Province of Limburg [10] reviewed the early studies and confirmed their conclusions, and placed these within our much-improved knowledge of the geology of The Netherlands [13,16,20] and of the Campine basin in Belgium on the other side of the Maas river [21].

Much of the existing geologic information on the Dinantian is currently being reviewed for the purpose of geothermal energy exploration [22–24], $CO_2$ sequestration [25], and for the mitigation of consequences of coal mine-water rise [26]. The conclusions of these studies are all consistent with those of the OPAC study.

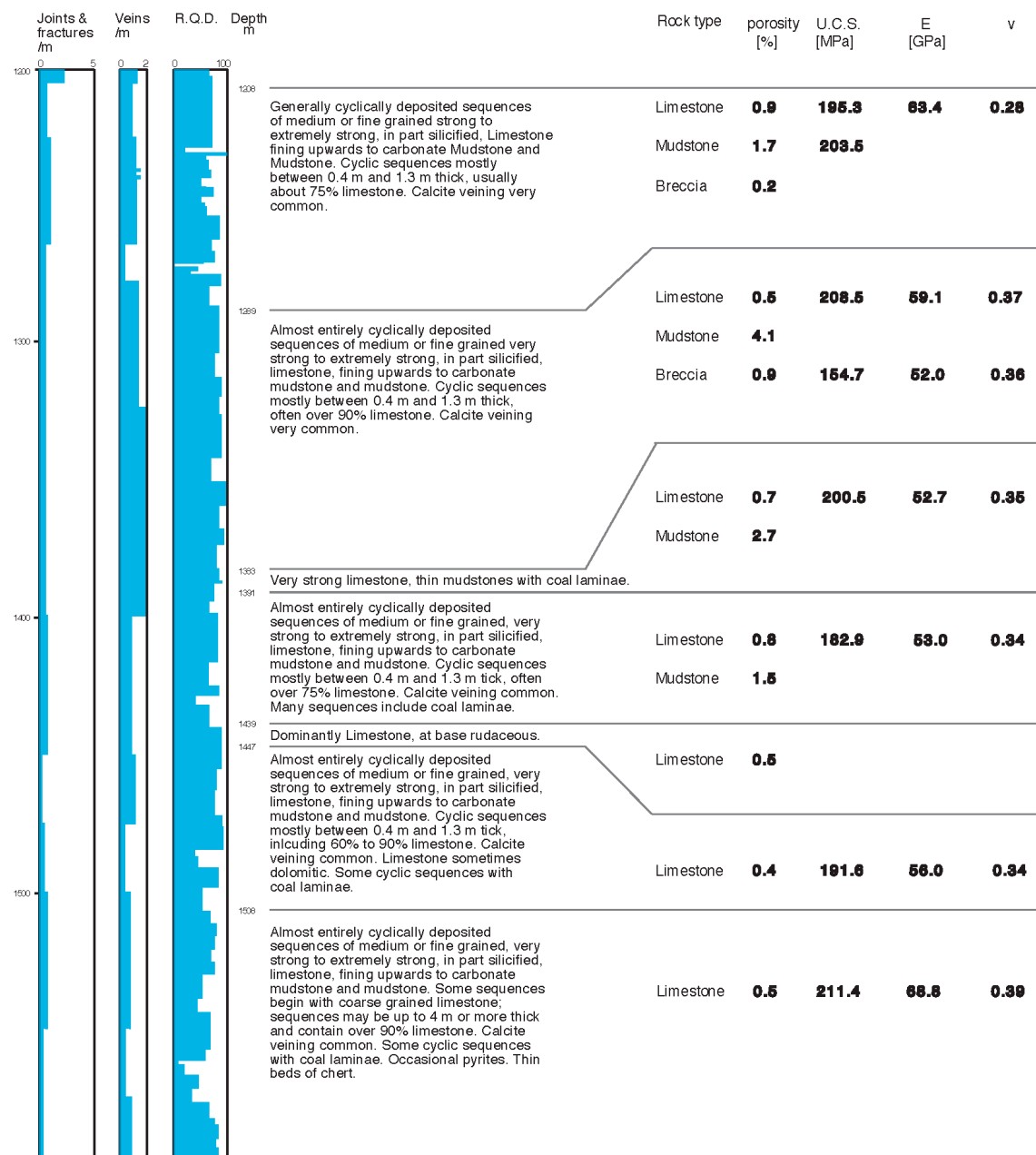

**Figure 4.** An example from the GVK-1 geotechnical database, showing in a graph the density of fractures, the density of veins (healed fractures), and rock quality (R.Q.D.) in the lower part of the core, together with average values of geotechnical measurements such as porosity (*n*), unconfined compressive strength (UCS), and Young's modulus (E).

All this makes clear that a mechanically very strong rock formation of approximately 700 m thickness is present in the area surrounding the Geverik-1 borehole, below 1000 m. The risk of not finding suitable rock at the required depth for construction of the lower reservoir is therefore very low. Using the much improved geophysical, structural modelling, and geomechanics methods, which have been developed in the past 30 years, it will now be possible to model and predict the geometry and properties of the minor fault and fracture systems in the subsurface, and optimize the position and layout of the lower reservoir cost of construction. An interesting advantage of O-PAC in comparison with conventional tunneling projects is the greater flexibility in choosing the exact location, which allows optimizing the design with respect to fault and fracture systems.

The next (development) phase can therefore build on this rich and comprehensive base of data. It will combine and integrate critical review and reprocessing of legacy data with a modern structural and geomechanical model. It will use high-resolution 3D seismic imaging to detect small faults at the limits of seismic resolution, and map subtle relay structures. Deviated drilling optimized to quantify the orientation and properties of the fracture systems using state-of-the-art image logs and cores, measurement of rock properties, and the stress field and fluid flow will create a mechanical earth model [27] which will be the base of an optimized design of the subsurface structures, using modern concepts of hydraulics [28].

Below we will briefly discuss some critical subsurface aspects of O-PAC, such as subsurface integrity, risk of soil subsidence, and the interaction with natural earthquakes. This is an initial assessment that, in the development phase, will be supported and validated by site-specific geological and geotechnical analyses, and subsurface designs.

### 3.1. Risk of Subsidence

The underground structures of O-PAC will be built in a stable, competent, impermeable rock; limestones with strength from strong (UCS > 100 MPa) to extremely strong (Figure 4). This is more than five times the total vertical stress at the relevant depth, clearly suitable for stable excavations. The 3D seismic investigations and wells drilled in the development phase will validate a subsurface volume with minimum sedimentary and tectonic disturbance, so as to minimize the need for rock support. The site investigations will optimize the location of a rock volume which is minimally impacted by faults, fractures, and (unlikely) karst. Geomechanical modelling will validate the very small movements during construction and help avoid significant reactivation of existing faults [29].

During operation of O-PAC, the potential energy will be stored in the upper reservoir and the lower reservoir will operate at close to atmospheric pressure, with a free water surface at 1400 m depth (Figure 1). This means that there will be no large fluid pressure and stress changes in the subsurface during operation of O-PAC. In such a case, because of the high elastic moduli of the rocks around the tunnels, very little, if any, subsidence is expected over the lifetime of the O-PAC facility. It should be emphasized that O-PAC will be a subsurface structure which is designed to be reliably stable over the very long-term operation of the facility, best comparable with tunnel construction in the Alps. Therefore, significant induced seismicity is not expected during operation.

### 3.2. Possible Damage by Earthquakes

The O-PAC will be built in an era of natural seismicity, but at sufficient distance to the active faults. It is well known that underground structures are affected less by earthquakes than structures at the surface. In very large (M > 6) earthquakes, underground structures may develop some damage but much less than surface structures. Damage tends to decrease with increasing overburden depth. Underground facilities constructed in soils tend to have more damage compared to those constructed in rock [30,31].

The main reason for this is that there are two kinds of seismic waves, body waves and surface waves. Surface waves only develop near the Earth's surface and play a negligible role in the subsurface. Seismic waves have a typical wavelength of a few hundred meters. Strains to underground constructions are dependent on the critical dimensions of the facility (cavern diameter and height, tunnel cross section) in relation to the wavelength. These critical dimensions are much smaller than the relevant wavelengths, so that, in most cases, the wave just passes through the structure, often unnoticed.

Underground structures are confined by the surrounding rock and its support, but surface structures are free to respond horizontally and develop much stronger motions. In more detail, seismic design for underground structures is done in terms of the deformations and strains imposed on the structure, while surface structures are designed for the inertial forces caused by ground accelerations [30,31].

In summary, for earthquakes even considerably stronger than the present level of seismicity in Limburg, damage to O-PAC subsurface infrastructure is very likely to be negligible, with adequate design of the machines. During the construction of tunnels there are rare cases when redistribution of stress by the excavation leads to small seismic events [29]. In the case of O-PAC the risk of this can be minimized during the design phase. Some rock bursts are normal during tunnel excavation, and can be handled by carefully adapted blasting and support.

### 3.3. Interaction with Groundwater

Interaction with groundwater in the subsurface is going to be minimal because, by their design, underground U-PHS reservoirs are isolated from the groundwater system. Fluid pressure in the subsurface reservoir will remain much lower than the fluid pressure in the surrounding of the underground excavations. In the construction phase, the natural fractures crossed by the tunnels will be stabilized for the long term with proven grouting methods, based on experience with much more challenging conditions (compare with the Gotthard Basis Tunnel).

The relatively small thermal and geochemical effects of the process water are summarized in some detail by Huynen [3].

The shafts, which have to pass along upper, permeable, and saturated layers, are to be constructed by means of ground freezing or adequate construction elements. Therefore, no lowering of the water table shall occur and subsidence caused by this will be minimal.

### 4. Costs and Benefits—Certainty and Uncertainty

The low subsurface risk in combination with the fact that all technical components of O-PAC are proven technology translates into a cost estimate for the capital project which is relatively certain, subject to a 15% contingency, as usual for this type of project. The income that can be generated from O-PAC after its completion (circa five years after the date of the final investment decision) for the subsequent decades is quite uncertain, however. In this section we assess costs and present our best estimate for the future revenue stream (the benefits). The generic aspects of the analysis in this section are relevant for any large-scale (U)PHS. But since the analysis makes use of the particulars of the O-PAC project, we refer to it by its project name, without implying a loss of generality.

### 4.1. O-PAC in the Power System and in the Electricity Market

The one thing that is certain about the power market is that in the coming decades it will show a growing volatility due to an ever-larger share of intermittent renewables in the power system. Arbitrage by means of energy storage will become a profitable activity once the volatility exceeds a certain level. One would anticipate that the price difference between periods of oversupply and of undersupply will be sufficient to merit the investment in electricity storage.

This is based, however, on a first-order approximation assuming that arbitrage does not influence market prices. However, when the arbitrage operation becomes larger, the low prices of the oversupply periods will increase by the extra demand of the arbitrage operation, and in the same way the price peaks of the undersupply periods will be lowered by the supply from the storage facility. This dampening of the volatility of the price of electricity reduces the income of a storage facility, especially a large one, but confers an advantage to society. Peak prices will be lower, and less electricity will be wasted in case of an over-supply by renewable sources.

The design of O-PAC is for a power rating of 1.4 GW and a useful storage capacity of 8.4 GWh. A full power cycle lasts roughly fourteen hours; eight charging and six discharging. This makes O-PAC ideally suited to cover and mitigate diurnal supply-demand mismatches.

If arbitrage is the main function of O-PAC and its chief source of revenue, a second additional source of income stems from O-PAC's ability to offer frequency control services. Within this class of services there is first, frequency containment reserve power (FCR). The ability to provide extra power to stabilize the net frequency if suddenly needed is of obvious value and commands a price. The payment

is for the guaranteed power, not for the energy supplied. The income derived from FCR is therefore additional to the arbitrage income. A second type of services is automatic frequency restoration reserves (aFRR), where the transmission system operator (TSO) invites bids to help support the net frequency. Manual frequency restoration reserves (mFRR), finally, come in two flavors, reserve power and incident reserve. Each is meant to restore a specific instability in the net.

Finally, O-PAC is the only facility that can provide a green black start after a power outage.

### 4.2. Costs—Initial Investment, Financing and Operation and Maintenance

While our focus in this section is mostly income, we merely summarize the project cost data for the "turnkey project", i.e., everything from development through to construction and commissioning, and including financing, permitting, etc. Table 1 lists these costs according to the latest estimates from the O-PAC consortium [32]. The numbers are close to those given in the recent monograph by one the authors [3]. A contingency of 15% is common for projects at the tendering/bidding stage, where O-PAC is at [33]. While 15% is low for public works, the value is justified because at the surface the project is not passing horizontally through public space or private property, and it penetrates vertically in the extremely well-researched subsoil (Section 3) that is owned by the project itself. Second, the O-PAC project is managed by private investors not by public authorities.

**Table 1.** Construction and other costs related to the O-PAC investment.

| O-PAC Investment Costs | | M€ |
| --- | --- | --- |
| Project development costs | | 55 |
| Total construction costs | | 1769 |
| *Vertical shafts* | 250 | |
| *Shaft installations* | 102 | |
| *Subterranean spaces (e.g., cooling, ventilation)* | 83 | |
| *Civil Engineering works* | 709 | |
| *Hydro- and electric machinery* | 324 | |
| *Electrical installations* | 33 | |
| *Above ground construction* | 37 | |
| *Contingency (15%)* | 231 | |
| Interest during construction | | 140 |
| Legal and financial fees | | 59 |
| Total investment (CAPEX) | | 2023 |

Above, we alluded to the fact that O-PAC needs government support and will most likely be executed in public–private partnership. In this context it is important to note that the construction and operation of O-PAC has a significant beneficial macro-economic effect for the surrounding Meuse-Rhine Euroregion. In an extensive study, Maks and Oude Wansink estimated the macro impact on the region. As an indication, the employment effect of the six-year construction period and the subsequent fifty-year operational life amount to circa 66,000 man-years in total [34].

On the cost-side, the financial viability of the O-PAC project is as much determined by the weighted cost of capital (WACC) as by the total capital expenditure (CAPEX). The WACC is, as the name implies, the weighted average of the required rates of return on the equity and the debt, which together fund the investment. Here we are assuming a moderate leverage, where a third of the project is financed by equity and two thirds by debt (bonds). On the necessary assumption that O-PAC is fundamentally a low-risk investment (of which more below, Section 5) a 5% return for equity investors is plausible. For debt we assume 3% (a conservative number at the time of writing when rates hover around 1%). This gives a WACC of 3.67%. With an economic and technical lifetime of 50 years (appropriate both for civil works and hydropower projects), this translates into an annuity of 4.39% of 2023 M€, or 88.8 M€ per year.

In addition, the operation and maintenance costs of O-PAC need to be factored in. Based on hydro power plant experience, this is estimated at 1.1% of the investment, i.e., 19.8 M€ per year (as an illustration it can be mentioned, that the O-PAC-consortium received a bid by suppliers to maintain the entire installation for only 10 M€ million/year).

Thus, the project will break even if a net revenue of 110 M€ per year can be guaranteed.

### 4.3. Base-Case Revenue for O-PAC—(i) Arbitrage

The future arbitrage income of O-PAC depends on the volatility of the future power market. This in turn is a derivative of the merit order formed by the future installed base of intermittent renewables, the aggregate dispatchable power production capacity and demand. The future installed base (both intermittent and dispatchable) is the outcome of government mandates, incentives, and subsidies on investor responses to these over a great number of years, the income is essentially unpredictable, something we return to in the next section.

Putting this epistemological obstacle aside, we can usefully say that arbitrage income is the product of the hours of extreme prices (low for buying and high for selling) times the average difference between these periods. There is no dearth of estimates for these. The 2050 outlook by Energy Brainpool for the EU [35] is a good example and is used here to illustrate the future business case.

As seen in Figure 5, it makes a projection for the number of hours of very high (>100 €/MWh) and very low (i.e., negative) power prices. The development over time is indicative of an electricity system that accommodates an ever-greater share of intermittent renewables. (The scenario is hardly aggressive in this respect, having "only" 36% of production across the EU-28 coming from solar and wind by 2050.)

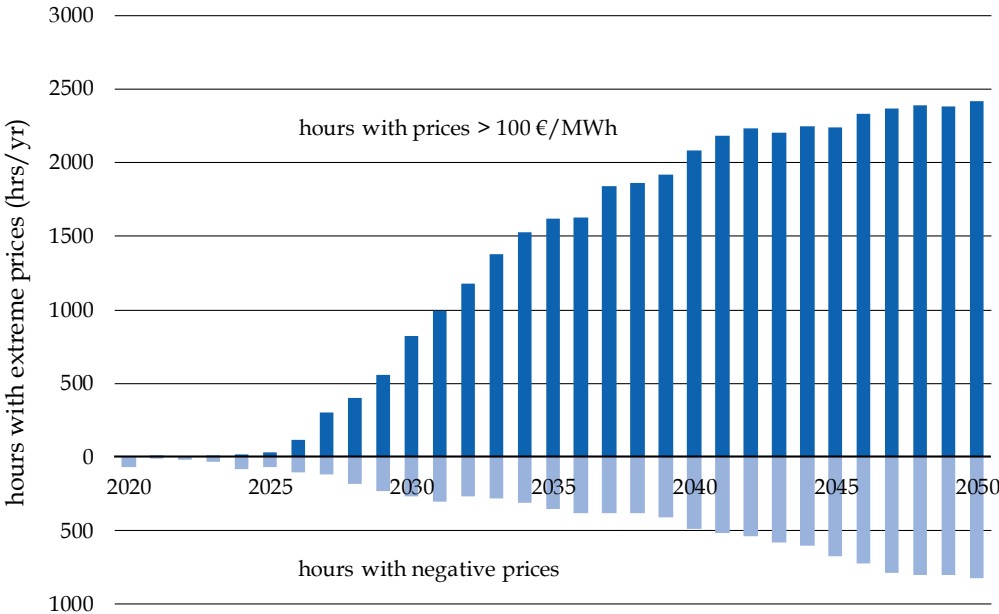

**Figure 5.** Projections for the annual hours of extreme prices in the EU28 according to a scenario by Energy Brainpool [35].

In such a system, momentary oversupply of electricity from wind and solar gives rise to negative prices, while the hours of peak prices are the consequence of phase-out of much of Europe's coal- and nuclear-based power generation capacity. This generic trend provides the incentive for investment in electricity storage (of any type) and is the basis for the revenue estimate for O-PAC.

Due to the losses in pumping and generation the round-trip efficiency of a pumped hydro storage system is 80%; a number proven in many operational hydropower plants in the world and guaranteed by equipment producers. This is considerably higher than other energy storage technologies like

compressed air (52%) or (green) hydrogen (36%), but comparable to battery storage (72–86%) [36]. This means that, from an economic point of view, all power storage requires a significant spread of power prices; batteries and (U)PHS the least, CAES a higher spread, and hydrogen so high a spread that its role in the future power system is less in power storage than the transformation of excess renewable power into clean (green) fuel.

In the simplest approximation, revenue can be estimated on the assumption that O-PAC delivers 1.4 GW during all high-price (>100 €/MWh) hours of the Energy Brainpool scenario [34], and consumes power to pump water up during all negative-price hours (as noted above, discharging at full capacity takes six hours; charging eight). Since there are less negative-price hours than needed to balance the plant, significant pumping will also be done at average prices. Assuming the high-price extreme to be between 125 and 200 €/MWh, negative prices between prices between −10 and −30 €/MWH and average ("normal") prices between 50 and 75 €/MWh, one gets a revenue range as shown in Figure 6.

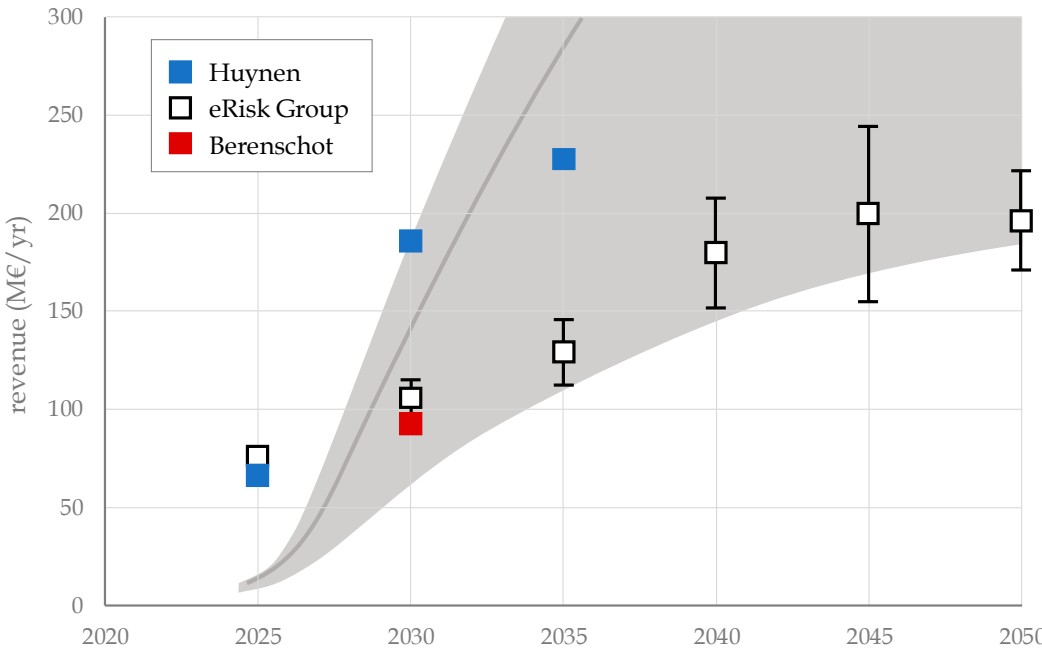

**Figure 6.** Estimates for the annual arbitrage revenue of O-PAC. The grey shaded area and grey line are based on the hours of extreme prices in Figure 5. The symbols represent more detailed revenue estimates by Huynen [3], eRisk Group [37], and Berenschot [38] and are discussed in the text.

These numbers can be compared with more detailed revenue estimates that have been specifically done for O-PAC, and that do not use a generic EU scenario, but are tailored to the anticipated Dutch and NW-European power market, with the specific renewables deployment targets of the Dutch government. The first is the 2018 monograph, by one of the authors [3], which uses the Energy Brainpool outlook to as a basis for revenue estimates. The second a more recent study by eRisk Group for the O-PAC consortium which gives an outlook to 2050 [37], and uses their proprietary PPSGen model [38]. The third is made by Berenschot, a consultancy firm, at the request of the Dutch Government [39]. It used Quintel's energy transition model to calculate future power demand [40], and subsequently generated hourly price forecasts and calculated revenues based on a valuation algorithm by KYOS [41]. All values from the reports are plotted in Figure 6.

By critical comparison, we may attribute the relatively high estimates by Huynen to the assumption of a 200 €/MWh price during the scarcity hours. Both the eRisk Group and the Berenschot studies explicitly take into account that O-PAC would have more than a marginal effect on the Dutch generating mix, and will therefore experience an electricity market that is less volatile due to O-PAC's active presence—the dampening effect described earlier. Next to income for the project owner, Berenschot

also gives an estimate of the societal (system) benefit of O-PAC of ca. 140 M€; a benefit that arises from the investment, but one that is not returned to the investor [42].

### 4.4. Base-Case Revenue for O-PAC—(ii) Reserve Power and Ancillary Services

If the main income of O-PAC comes from arbitrage, a not insignificant additional revenue stream may be generated by offering reserve power in the balancing market. The types of control have been identified at this introduction of this section. A rough estimate of additional income from balancing is given in Table 2, which is taken from Huynen [3], and based on a report by E-bridge [43].

**Table 2.** Estimated annual revenue from frequency control reserves [3].

| Service | Power (MW) | Price | Revenue (M€/y) |
|---|---|---|---|
| FCR | 100 | 2452 (€/MW·week) | 12.8 |
| aFRR | 150 | 7000 (€/MW·month) | 12.6 |
| mFRR | 150 | 1800 (€/MW·month) | 3.2 |
| Total FC revenue | 400 | | 28.6 |

One must take proper account of the fact that in exchange for this income 400 MW out of the 1400 MW of the installation has to be kept in reserve. Consequently only 1000 MW is available for arbitrage. This revenue is therefore only fully additional to the arbitrage revenue in the first years of operation, when the full arbitrage capacity is not yet used. Finally, some (unquantified) income may be generated by remunerated ancillary services: redispatch, black start, and reactive power.

### 4.5. Benefit/Cost Ratios from a Private and a Societal Perspective

In the benchmark year of 2030, the benefit to the O-PAC owner/operator is in excess of 100 M€ in all scenarios. The possibility of securing a significant income from frequency control, in case arbitrage income would fall short of expectations, reduces uncertainty. The costs were estimated at just under 110 M€, including 90 M€ for capital at a 3.67% WACC and a 50-year project life. This gives a private benefit-to-cost ratio of about one, making the project just viable, especially if one accounts for the fact that the arbitrage income has headroom in later years.

From a societal perspective the benefit is an estimated 140 M€ higher, due to the system benefit it delivers (vide supra). This brings the societal benefit-to-cost ration well over two, making it a very attractive proposition form this perspective.

This marked discrepancy between private and societal rewards merits further discussion, especially in relation to the non-technical, transition risks. To this we now turn.

## 5. A High-Level View on Risks and Rewards

From a private investor-perspective, O-PAC is a difficult proposition (which the O-PAC development team is experiencing). It combines a relatively long construction period (circa six years from the final investment decision to start of operation) with a high non-technical, i.e., political risk, namely that of evolving government monetary and energy policies (viz. interest rates, subsidies on renewables, corporate tax, $CO_2$ taxation, power market regulation, etc.), and an extremely long capital recovery time of 50 years. As we showed in Section 4, financial break-even operation is plausible at a WACC of 3.67%, the average of 67% debt financing at 3.0%, and 33% equity requiring a 5% return. This would be acceptable for a low-risk project, which O-PAC is not. Not, that is to say, from the investor perspective for all the reasons cited above. Owing to the above non-technical uncertainties and the ensuing risk profile, investors would seek a higher return than can reasonably be projected.

It is important to note that the uncertainty is largely due to government policies and how they direct and incentivize the rebuilding of the power system, from its fossil-dominated present state to a largely renewables-based one. Government to a large degree owns this risk, and it can mitigate it. It (or rather the public at large) is also the beneficiary of the lower power system cost that O-PAC

enables, which translates to a monetary benefit that is at least as high as the revenue that flows back to O-PAC's investors (Section 4).

This leaves us to prove that there are no alternatives to O-PAC, and (U)PHS more generally, that can deliver the same benefits more cheaply, and/or without government support. The main (classes of) contenders are battery storage and hydrogen. We will review their merits in comparison to O-PAC briefly.

Batteries will play an increasing role in the power system, but their main use is in frequency control, and they are fundamentally smaller, and more costly than (U)PHS. Lazard, which keeps track of storage projects, puts the unsubsidized levelized cost of storage (LCOS) of the largest projects (100 MW, 400 MWh) for the wholesale market at 165–305 $/MWh [44]. The LCOS for O-PAC has been estimated at 48 and 41 €/MWh by Huynen [3] and Berenschot [39], respectively. This shows that battery projects are more than an order or magnitude smaller than O-PAC, and more expensive. To be commercially viable, batteries require very a high volatility in the power market, driving their application effectively to frequency control.

Great hope is vested in the vehicle-to-grid (V2G) use of the electric vehicle (EV) fleet that is being built up. Huynen showed for The Netherlands, in a scenario with one million EVs, with 50–100 kWh batteries, the effective storage made available for arbitrage might be comparable to that of O-PAC. The cost is difficult to assess as the investor, i.e., the vehicle owner, bought the battery for the purpose of vehicle range, and V2G would be an additional (bonus) income. Rationally, the bottom price that the vehicle owner would set is such that the V2G revenue at least recompenses for the cost of additional battery degradation. This was behind Tesla's stance against their vehicles being used for V2G, a position they only very recently reversed [45]. When the Dutch EV fleet reaches the million mark, and if their owner/users are inclined to operate them in V2G mode, it will only be complementary to what O-PAC can deliver. The demand for electricity storage by 2030 can certainly accommodate both side by side, and where the storage of the vehicle fleet is dispersed and fundamentally uncertain, O-PAC is central and always available. From a portfolio perspective, the combination is ideal.

Hydrogen is a fundamentally different storage proposition. Its low round-trip efficiency makes it fundamentally unattractive for electricity storage. (Green) hydrogen production through water electrolysis is likely to play a major role in the future energy system, but primally for the offtake of renewables production when the production structurally exceeds the primary electricity demand (i.e., the demand excluding that for storage and/or electrolysis). It thereby enables an "overbuilding" of intermittent renewables, so that the periods of oversupply become more frequent and those of undersupply less so. This would affect and ultimately reverse the assumptions of the Energy Brainpool scenario used in Section 4 and plotted in Figure 5, which takes undersupply to be dominant. Of course, green hydrogen can ultimately be used as a fuel replacing natural gas in (back-up) power generation. However, it will not directly compete with O-PAC and battery storage. Those have superior properties for diurnal storage. Hydrogen kicks in only as seasonal storage and as a source of green fuel for transport, and is thereby (again) complementary to O-PAC.

Should therefore electrolysis and green hydrogen production take off at scale, its main influence on O-PAC is less on the utility of O-PAC for system balancing, which it does not much effect, but rather on the balance between the hours of over- and undersupply by renewables, which it will shift towards oversupply, which may reduce the arbitrage revenue in the long term.

Having looked at these two classes of alternatives, we conclude that O-PAC, and by extension (U)PHS, is an attractive addition to the portfolio of storage options that must be brought to development in order not to stall the massive deployment of solar and wind necessary to meet climate-related emission targets. This is a formulation that is tailored to the Dutch situation which is virgin territory for grid-scale storage. Most countries have PHS as the default storage option. For those countries, the above analysis proves that PHS is not a relic of the past, superseded by batteries or hydrogen, but is more relevant than ever. This explains that across the world, the list of new PHS projects and expansions is growing [46].

For The Netherlands and O-PAC, the absence of a "tradition" in hydropower storage investments, creates a hurdle that is yet to be overcome. Much of the PHS that we have in the world today, which is pretty much all the storage, was built in an era of regulated utilities. In this paper we have shown that O-PAC, and by extension (U)PHS in general, is a sound investment in public infrastructure and eminently helpful in sustaining the case for the continued build-out of renewable power. While at the same time one that is insufficiently attractive to attract private capital without government guarantees to cover the transition risk. In today's deregulated and unbundled electricity market, such guarantees, let alone a public private partnership, go against the ruling idea that "the market will deliver" the solution. Even if it is evident that (U)PHS will not be built under these conditions, hope is put on batteries and hydrogen to be different; a hope that can be sustained in conjunction with the belief that costs will come down and the investment case will arise soon enough. This denies the fundamental issue that all storage competes with itself.

Findings in a recent paper co-authored by one of the authors, "Why fully liberalised electricity markets will fail to meet deep decarbonisation targets even with strong carbon pricing" [47], underscore the generic nature of the investment in electricity storage. Based on a generic agent-based model of investor behavior, the paper concludes that "when renewables deeply penetrate the electricity mix, (the market) does not give the right mix of investment signals to investors to invest in an efficient mix of renewables and storage assets." In their model, this inability for the "market to deliver" was shown to be especially strong for investments with a long lead time, and when the generation mix is in transition towards deep penetration of renewables.

## 6. Conclusions—The Path to Investment in U-PHS in The Netherlands

In this paper we laid out the case for O-PAC. It is ready to be built. The geology and geography of Limburg are favorable. It can be built and, in all fairness, it should be built as the time is right. As a public utility project it is very attractive, with a high societal benefit/cost ratio; we calculated it to be around two, and did not even consider the positive macro-economic multiplier effects of the construction phase.

We have shown that the risk profile of O-PAC is dominated by the non-technical risk of how the energy transition will be managed; how much intermittent renewables will be built and by what date, and if and how further incentives will be given for V2G of the electric vehicle fleet and to green hydrogen production. High risk in itself is not an obstacle to private investment, but only if the downside risk is offset by a significant upside risk. However, as we have seen, it is in the nature of electricity storage that there is rather little of that; massive storage kills the (commercial) business case for arbitrage, even as it delivers a very tangible benefit to society, by the reduction of the overall (electricity) system cost.

This explains that, when left to the market, the technologies and business models that attract most business interest are those for which arbitrage revenue is a side benefit and not the primary investment purpose. The foremost example is V2G. An EV buyer primarily invests (if that is the word) in a car, which comes with a battery that gives the car its range. V2G arbitrage income is merely nice to have, and not a financial driver. Similarly, utility-scale battery systems are primarily seen in the frequency control market, not in arbitrage. If and when hydrogen production through electrolysis takes off, its driving force will not be arbitrage either, but (flexible) power use to produce a fuel.

Thus, the market simply does not incentivize investment into massive, dedicated grid-scale storage, whether (U-)PHS or other. If the nation wants to reap the system benefit of it—a considerable benefit, as we have seen, it has no choice but to take away the non-technical risk for the investor. Only when the risk is low, will investors step in to reap the modest project returns that grid-scale storage can generate. The good thing is that government, to a good degree, "owns" that risk and can control it by orchestrating the timing of investments necessary to deliver a cost-efficient power system based on intermittent renewables and appropriate storage.

**Author Contributions:** Conceptualization, J.M.H.H. and G.J.K.; formal analysis, J.L.U. (Section 3), J.M.H.H.; T.A.; H.V. and G.J.K. (Section 4); investigation, J.M.H.H. and T.A. (Section 2); data curation, T.A. and J.L.U.; writing—original draft preparation, all; writing—review and editing, G.J.K.; visualization, J.L.U. and G.J.K. All authors have read and agreed to the published version of the manuscript.

**Funding:** This research received no external funding.

**Acknowledgments:** The authors acknowledge Bert den Ouden of Berenschot for discussion of their analysis, in particular of the societal benefit calculations.

**Conflicts of Interest:** J.M.H.H. is an entrepreneur who is the initiator of the O-PAC project. T.A. is member of the core team responsible for O-PAC project development. J.L.U., at Geostructures, is an advisor of O-PAC.

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
