# Peer review of "Risk Mitigation and Investability of a U-PHS Project in The Netherlands"

_energies, doi:10.3390/en13195072_

Round 1

Reviewer 1 Report

The paper describes a 1.4 GW 8 GWH U-PHS project in the Netherlands which has received attention since the 1980s. The article is mainly a discussion around the merits of the system, discussing the history, the geology and presenting some information about the economics. Overall I think the article is interesting and will be of interest to the energies readership. However, while I appreciate that the authors are supportive-of/involved-with the project I think that in order to be published in a scientific journal some nuance must be added, so that the article reads a little less like an opinion piece and more like an article reviewing evidence. More details on the sources of the income streams must be given, including where the rewards for the reserve services come from. This would also give the impression that the conclusions were better supported by the arguments presented.

The abstract gives the distinct impression that one of the reasons for a lack of investment in UPHS is the Netherlands lack of experience with PHS. However, no country has ever built a large-scale UPHS plant, despite many having experience with PHS there are only proposed project. Therefore this sentence needs changed to reflect this.

Lines 57 "As an 57 infrastructure investment however, the project has a good monetary return at the system level" is not fully backed up by the data. "System level" is not explained here - the costs of running the electricity system are not discussed in the paper although it is thought that storage generally reduces the running costs of the electricity system.

The authors assert that the 2,528 Million guilders investment as calculated in 1989 is equivalent to approx 2 billion euro today. This seems low - so there is less than a factor of 2 increase in cost over 30 years? What about inflation??

Paragraph starting 158 - some nuance should be added here. Of course the components are proven technology but not for UPHS.

Section 4: Costs and Benefits

Where does the 15% contingency come from some reference should be provided. Furthermore, some discussion of the fact that the costs and construction periods of large infrastructure projects often more than double. For example the big dig in Boston went from 2.8 billion to over 8 billion dollars and the crossrail project in London has already gone from £14.8bn to £18.25bn.

Line 366 - typo - hoover ---> hover

Line 367 - the initial economic lifetime of most plants is significantly shorter than 50 years, this should be recognised.

One of the major risks with arbitrage is illustrated by Figure 5 - Initially there are relatively low numbers of prices with >100 euro/MWh. Therefore unavailability or unplanned outage over a few hours would expose the operator to a large revenue loss. Some discussion of this must be given, especially since these price spikes would need to be forecast.

Table 2 is missing information about where the prices come from and the exact assumptions. Obviously these are so important that they should be included the discussion with more information about the exact assumptions and information about why the assumed prices were chosen (references to current plants providing reserve services).

Line 478 - the levelised cost of batteries is provided but there is no comparison to a levelised cost from this project.

Line 494 - this sentence should be changed. As the authors recognise later in the paragraph, if there is a surplus of renewable generation then storage round trip efficiency is less important. Hydrogen is of course still a very expensive option due to the costs of electrolysers.

Line 501 - This economic analysis only uses the Energy Brainpool scenario, this should be mentioned as a key limitation earlier.

In general, the risks with this type of project are mainly regulatory, and with the changing nature of the energy sector these are difficult to navigate for an infrastructure asset where long term contracts are not achieved. The authors also underplay the importance of the market structure - there have been very few major energy storage plants commissioned in liberalised electricity markets [1], most have been commissioned in a monopoly market of some kind. This should be further discussed.

Overall, the project is very sensitive to the regulatory environment in which it may find itself. The paper concludes by suggesting that the government may own the risk associated with the project allowing investors to step in and reap the modest project returns. However given that the government owns the risk, some discussion of the merits of keeping the returns public should also be given. This is especially pertinent given that the government would be able to raise the necessary project capital at a lower cost than a private investor. The authors already discuss the market failure in relation to large-scale energy storage.

[1] Barbour, Edward, et al. "A review of pumped hydro energy storage development in significant international electricity markets." Renewable and Sustainable Energy Reviews 61 (2016): 421-432.

Author Response

we have provided replies to both reviewers in one file which is attached.

Reviewer 2 Report

The paper presents with an holistic perspective investigating the feasibility of underground pumped storage hydro in the Netherlands (a country with very limited natural elevations). This gives an interesting overview of the potential of such a technology, as well as the obstacles for its practical implementation (regarding all aspects of the project). The contribution is very well written, well structured, and is of interest to highlight the viability of this solution (as pumped hydro may be wrongfully associated with areas with overground height differences). It also presents comparison with rival technologies, and show promising complementarity in the future energy mix. The risk analysis is also very insightful. You can find hereunder some suggestions/questions that may deserve further discussions.  

The arbitrage profit seems to be computed only with extreme prices (as reported in lines 408-415 based on Fig. 5). This is a very conservative solution, especially in the short-run since even the current (moderate) spreads between hourly prices leave room for additional profits. What is the main driver for this choice?

In page 12, what is the rationale behind the choice of 400 MW (out of the 1,400 MW) for the provision of reserves, to leave 1,000 MW for arbitrage in energy markets? In that regard, it is worth mentioning that, in actual operation, a daily optimization has to be performed (where, e.g. more reserves may be offered in days with low discrepancies in energy prices).   In section 4.4, potential revenues for the provision of reserves are computed. However, it seems that only the profit for the reservation (availability) of power is considered. The revenues would be even greater (for aFRR and mFRR) when profit for the actual provision of reserve (in terms of energy) is considered (as there may be a price spread between upward and downward products). It may be worthwhile to further comment on this assumption in a revised version of the paper.   Due to the increased variability of the electricity generation (and hence of the electricity prices), the idea to exploit underground PHS is (re-)emerging. In particular, small-scaled solutions exploiting natural underground basins (in order to significantly reduce investment costs) have also shown economic potential, see [A]-[B] where U-PHS generates daily profit by jointly participating in energy and reserve markets. It may be interesting to add these findings to complement the state of the art in the potential of the technology. Moreover, such solutions may be good 'proof-of-concept’ (pilots) leading to the implementation of large-scale infrastructures. [A] Toubeau J. , De Grève Z., Goderniaux P. , Vallée F. and Bruninx K., "Chance-Constrained Scheduling of Underground Pumped Hydro Energy Storage in Presence of Model Uncertainties," in IEEE Transactions on Sustainable Energy, vol. 11, no. 3, pp. 1516-1527, July 2020, doi: 10.1109/TSTE.2019.2929687. [B] Toubeau J. , et. al, « Non-linear hybrid approach for the scheduling of merchant underground pumped hydro energy storage, » IET Generation, Transmission & Distribution, vol. 13, no. 21, pp. 4798-4808, 2019.   Due to its size (1,400 m of height difference between both reservoirs), it seems that the O-PAC is not significantly affected by the head effect (which incurs dynamic variations of the safe operating range, thus complicating the daily operation strategy). This added value wrt. small sized underground PHS is worth mentioning in the manuscript.   In line 671, the number [42] is missing.

Author Response

we have replied to both reviewers in one file which is attached.

Round 2

Reviewer 1 Report

After a first round of reviews, the authors have added more evidence to support their claims which has improved the paper. While I still have reservations as detailed below, in principle I am happy for the paper to be published to stimulate further discussion in the area.

Reservations

In general, I feel that the uncertainty associated with the project is significantly understated by the authors.

  1. The contingency costs of 15% are taken from reference 33, which as I understand attributes costs based on a maturity matrix. This is a first of a kind plant. What maturity level has been assumed? I feel the authors should introduce 80% confidence levels and present the uncertainty on the returns.
  2. There is significant uncertainty in the very high and very low electricity prices required for the arbitrage revenue. Many future changes could significantly diminish this spread, including levels of EU interconnection, demand levels, consumer behaviour changes, demand response, etc. A description of the key features of the price scenarios used is missing in this regard. There are also question marks about the ability to accurately forecast these high/low price hours and this is not discussed.

Author Response

We thank the reviewer for his second round of comments and we are very happy with his support for publication of the paper.

The two points that the referee raises are restatements of concerns the referee also expressed in the first round. Both of them have to do with the financial uncertainties of the project: the first question relates to uncertainty in costs; the second to uncertainty in revenue.

In the first question the reviewer challenges the 15% contingency on the basis that the project is first of a kind. In the first review round the same (?) reviewer challenged the 15% contingency on the basis that “the costs and construction periods of large infrastructure projects often more than double”. And, also in the first round of review, “proven technology” was challenged. We responded to both in our earlier response to the reviewers.

On the issue of “proven technology” we pointed out that all technical components of O-PAC are proven technology, including working at great depth, albeit not in the Netherlands. But, as we pointed out in our response to the question on contingency, the subsoil at the project location is extreme well researched.

On the issue of 15% contingency being low, we have now amended the text to include the justification that we gave in response to the reviewer in the first round, but which had then not been added to the text. We have added two sentences on starting on line 347 (new text in italics, below):

A contingency of 15% is common for projects at the tendering/bidding where O-PAC is at [33]. While 15% is low for public works, the value is justified because at the surface the project is not passing horizontally through public space or private property, and it penetrates vertically in the extremely well-researched subsoil (section 3) that is owned by the project itself. Secondly, the O-PAC project is managed by private investors not by public authorities.

As to the second question: the reviewer rightly points out that there is very significant uncertainty in future electricity prices and hence in the revenue. We fully agree. If fact we say so explicitly in the text. The opening paragraph of section 4.3 is entirely devoted to this, ending with “… the income is essentially unpredictable” (lines 379-80 in the 2nd revision of the manuscript). It comes back in the second paragraph of 5 section (“It is important to note…”, line 471-6) and leads to final conclusion that in order to make the project investable, government must step in to cover the risk associated with future power market development. So the notion of revenue uncertainty is very much woven into the fabric of our argument and in the text as is.

We trust that this response and clarification is satisfactory.